# Atlantic Salmon (*Salmo salar*) Transfer to Seawater by Gradual Salinity Changes Exhibited an Increase in The Intestinal Microbial Abundance and Richness

**DOI:** 10.3390/microorganisms11010076

**Published:** 2022-12-27

**Authors:** María F. Morales-Rivera, Diego Valenzuela-Miranda, Gustavo Nuñez-Acuña, Bárbara P. Benavente, Cristian Gallardo-Escárate, Valentina Valenzuela-Muñoz

**Affiliations:** 1Interdisciplinary Center for Aquaculture Research (INCAR), University of Concepción, Concepcion 4030000, Chile; 2Laboratory of Biotechnology and Aquatic Genomics, Department of Oceanography, University of Concepción, Concepcion 4030000, Chile; 3Centro de Biotecnología, Universidad de Concepción, Concepcion 4030000, Chile

**Keywords:** Atlantic salmon, parr-smolt transformation, seawater transfer, intestinal microbiota, nanopore sequencing

## Abstract

The host’s physiological history and environment determine the microbiome structure. In that sense, the strategy used for the salmon transfer to seawater after parr-smolt transformation may influence the Atlantic salmon’s intestinal microbiota. Therefore, this study aimed to explore the diversity and abundance of the Atlantic salmon intestinal microbiota and metagenome functional prediction during seawater transfer under three treatments. One group was exposed to gradual salinity change (GSC), the other to salinity shock (SS), and the third was fed with a functional diet (FD) before the seawater (SW) transfer. The microbial profile was assessed through full-16S rRNA gene sequencing using the Nanopore platform. In addition, metagenome functional prediction was performed using PICRUSt2. The results showed an influence of salinity changes on Atlantic salmon gut microbiota richness, diversity, and taxonomic composition. The findings reveal that GSC and the FD increased the Atlantic salmon smolt microbiota diversity, suggesting a positive association between the intestinal microbial community and fish health during seawater transfer. The reported knowledge can be applied to surveil the microbiome in smolt fish production, improving the performance of Atlantic salmon to seawater transfer.

## 1. Introduction

Over the past years, it has gained consensus that fish health and welfare are highly influenced by their microbiota [1]. The microbiota of marine fish can play pivotal roles in gut development, metabolism, and immune responses. Understanding how microbiomes are modified is important to understand the host-microbe interactions and their influence on host health [2]. Therefore, different research efforts have been focused on understanding the factors that shape gut microbiota in marine fish to promote healthy organisms and welfare in aquaculture production [3].

The fish microbial community has mainly been explored by molecular methods based on sequencing technologies. Herein, the sequencing of the hypervariable regions from V1 to V9 of the 16S rRNA gene was used to study the microbial diversity of marine environments [4,5,6,7]. There is no consensus on which regions should be used to assess the fish microbiota. Thus, it is possible to find studies that use different regions of the 16S rRNA gene for microbiota identification [8,9,10]. However, it is known that the sequencing part of hypervariable regions may affect the taxonomic diversity estimation [11,12,13]. In addition, it has been observed that different 16S rRNA regions alone are not sufficient for precise estimates of richness and higher-level taxonomic classification [14,15]. In recent years, the sequencing of the full 16S rRNA gene with Nanopore technology (Oxford Nanopore Technologies, Oxford, UK) has been suggested as the best tool to identify microbiota for species and higher taxonomic levels [16,17,18,19].

The microbiota structure can be modulated by diverse factors, including host physiological history, diet, ecological factors, and environment, among others [20,21,22]. Regarding environmental factors, it has been evidenced that changes in salinity can influence the structure and diversity of fish gut microbiota [23,24]. In euryhaline fish, the transfer to SW triggers fish drink rate increase [2,25]. These adaptations stabilize osmotic pressure alteration in the gut, which causes changes in the bacterial community that composes the gut microbiota [2,10,26,27].

In Chile, Atlantic salmon (*Salmo salar*) is the preferred species of salmonids on farms, accounting for 72% of production [28]. During parr-smolt transformation, or smoltification, Atlantic salmon require molecular and physiological changes that allow the fish to change from FW to SW [25,29]. In natural conditions, salmonids pass through a salinity gradient from the river to the sea during smoltification [30]. However, in productive systems, smolts are often transferred from FW to SW directly, without considering the impact of a direct change in salinity in the salmon microbiota and health. A larger diversity and richness of the skin microbiota in SW has been reported when compared with the FW phase one and four weeks after SW migration [8]. Wang et al. [10] reported a high difference in the Atlantic salmon gut microbiota between fish in FW and fish two weeks after the SW transfer. Among their results, a high abundance of the genera *Lactobacillus* and *Photobacterium*, associated with FW, and *Lactobacillus* and *Mycoplasma* was observed after 22 weeks in SW. Furthermore, in recirculating aquaculture systems [31], gut microbial richness decrees one week after SW has been observed [9]. In addition, it has been reported that microbial community changes in Atlantic salmon during the SW transfer can induce the fish immune system to allow an infectious disease outbreak [31]. Thus, functional diets have been developed to enhance salmonids’ performance and health during smoltification [32].

Previously, we reported transcriptome variation between the Atlantic salmon intestine exposed to gradual salinity change and salinity shock [33]. The study reported a high number of transcripts differentially expressed in the intestinal tissue of salmon exposed to a gradual salinity change compared with fish exposed to salinity shock. Furthermore, a Gen Ontology (GO) enrichment analysis showed a high abundance of genes associated with metabolic processes, ion transmembrane transport, and immune response. Thus, it is possible to suggest changes in the Atlantic salmon intestine microbiota under different smoltification strategies. Therein, this study aimed to explore the intestine microbiota richness and abundance in Atlantic salmon smolts at genus and species level by full 16S rRNA gene Nanopore sequencing. We hypothesized that the different salinity conditions and the use of in-feed functional ingredients could modulate the microbiome community during parr-smolt transfer. Collectively, the findings can support novel strategies for smolt production, improving animal health and welfare in salmon aquaculture.

## 2. Materials and Methods

### 2.1. Experimental Design

The Atlantic salmon intestine samples for this study were taken from the assays reported previously for our group [33]. Briefly, freshwater (FW) smolts (60 ± 6.2 g) were obtained from a commercial farm (Hendrix Genetics Aquaculture, Villarrica, Chile) and transported to the Marine Biology Station, Universidad de Concepción, Dichato, Chile. Fish were maintained in ultraviolet-treated by single-pass flow-through tank systems (500 L, 30 fish/tank) on a 12:12 h light: dark cycle, with a dissolved oxygen level of 8.5 mg/L and pH = 8.0, fed daily with a regular diet of Cargill. After one month of acclimation, FW (control group) samples were taken. Then, the fish were separated into three experimental groups in triplicate. The first group of salmons, 30 smolts per tank in triplicate, was exposed to a gradual salinity change (GSC) by increasing FW’s salinity to SW. The gradient was set at 3 salinity points, changing 10 PSU per week for a month. Intestine samples were taken after a week of acclimation in each salinity point. Meanwhile, another group of 30 smolts per tank in triplicate was exposed to a salinity shock (SS) and a transfer from FW to 32 PSU on the same day as the GSC group. These two groups were fed a regular diet of Cargill. The third group of 30 smolts per tank in triplicate was fed for a month with a functional diet (FD) of SuperSmolt Feed Only, Stim company (unsolicited study), and transferred to 32 PSU at the same time as the other groups. Fish intestine samples of each group were taken after a week of acclimation at 32 PSU. Furthermore, the salmon condition to SW transfer was evaluated by immune histochemistry analyses performed by the VEHICE company, Chile. In addition, RT-qPCR expression analysis of ATPase-α and ATPase-β was determined using the comparative ΔΔCt relative expression analysis. The primers and qPCR conditions were similar to those previously described for our group [33]. Animal procedures were carried out under the guidelines approved by the Ethics Committee of the University of Concepción. The experimental design considered the Three Rs (3Rs) guidelines for animal testing.

### 2.2. DNA Extraction and Full-16S rRNA Amplification

After the fish intestine digest was removed, intestinal tissues were thoroughly minced and then mixed with 1 mL of lysis buffer (10 mM Tris-HCl, 400 mM NaCl, 100 mM EDTA, 0.4% SDS, and 5 µL of 20 mg/mL Protease K, pH 8.0), after which they were thoroughly vortexed and incubated for 2 h at 37 °C. The samples were homogenized for 5 min at 24 Hz in an oscillating mill (MM200, Retsch) using 1.4 mm ceramic spheres, followed by the phenol:chloroform:isoamyl method. DNA integrity was assessed by 1% agarose gel electrophoresis after extraction. The DNA concentration was measured with a Nanodrop spectrophotometer. Five individuals per experimental group were pooled in a final concentration of 100 ng/µL. The full-16S rRNA gene was amplified using the primers 27 F 5′-AGAGTTTGATCCTGGCTCAG-3′ and 1492 R 5′-GGTTACCTTGTTACGACTT-3′ [34]. Fifteen µL Taq DNA polymerase LongAmp (New England Biolabs, Ipswich, MA, USA) was used for amplification, under the following conditions: 95 °C for 1 min, followed by 25 cycles of 95 °C, 56 °C for 30 sec, and 65 °C for 1 min, with a final extension at 65 °C for 5 min. Electrophoresis in agarose gel at 1.2% was used to assess PCR results.

### 2.3. Nanopore Library Synthesis and Sequencing

For Nanopore, the library was conducted in triplicate per each experimental group, using a full 16S rRNA PCR product of a pool of five individuals. The PCR products were purified in a ratio of 1:2 sample:beads using Agencourt AMPure XP beads (Beckman Coulter, Brea, CA, USA). The samples were placed in the magnetic stand for washing with 70% ethanol (freshly prepared). The beads were resuspended with molecular biology-grade water. The purified amplicon was quantified in the Qubit 4 fluorometer (Thermo Scientific, Waltham, MA, USA) and used as a template for library synthesis with the 16S Barcoding Kit (SQK-RAB204, Oxford Nanopore Technologies), following the manufacturer’s instructions for the 1D sequencing strategy. Briefly, a purified 16S amplicon was mixed with barcodes, and a PCR was done as described in the preceding stage, with a final reaction volume of 50 uL, using the LongAmp Taq Polymerase (New England Biolabs). The PCR product was incubated for 5 min at room temperature in a HulaMixer (Thermo Scientific) and cleaned with Agencourt AMPure XP beads. The purified product was eluted in 10 uL of elution buffer (10 mM Tris-HCl pH8.0 with 50 mM NaCl). The final concentration of the library was measured using a Qubit 4 fluorometer (Thermo Scientific), and libraries were tested using High Sensitivity D5000 ScreenTape (Agilent, Santa Clara, CA, USA), according to the manufacturer’s instructions, on a TapeStation Bioanalyzer 2100 (Agilent). Following Oxford Nanopore Technologies’ methodology, the libraries were pooled in multiplex mode and sequencing into the flowcell MK1 Spot-ON FLO-MIN107-R9. As an internal control, the DNA of a microbial mock community (Zymo Biomics Microbial Community Standard) was extracted and sequenced following the same described procedures, and the observed abundances of taxa were compared with their expected abundance. Sequencing efficiency was monitored through the software MinKNOW 2.0 (Oxford Nanopore Technologies).

### 2.4. Bioinformatic Analysis

The fast5 files generated were base-called using Guppy (version 3.2.2, Oxford Nanopore Technologies, Oxford, UK), and a filter step was applied to retain only sequences with a Q-score ≥ 7 (quality filter). The demultiplexing, primers, and barcode trimming was done with Porechop V0.2.4 [35]. BLASTN aligned cleaned sequences 16S NCBI taxonomy database in the NanoCLUST pipeline [36]. This included filtering by sequence size 1400–1800 bp and Q-score ≥ 9. Simpson’s 1-D and Shannon index were used to estimate alpha diversity, while Pielou’s index was used to estimate evenness. Microbial community structure was analyzed using Bray-Curtis dissimilarity based on taxon relative abundance data, and principal coordinate analysis (PCoA) was performed as a multivariate unsupervised data exploration. All indexes and PCoA were calculated using R’s “Vegan” package [37]. Variations of the intestinal microbiome were explored by plotting the relative abundance of phyla, orders, and genera in R statistical software [38]. PICRUSt2 software was used to predict metagenome function using the V3–V4 region obtained from the full16S rRNA sequences [39]. These regions were identified by an alignment between our sequencing data with universal primers [26] for V3–V4 in CLC program. The pathway levels were built using the MetaCyc database [40]. A relative abundance plot was generated using R. We used the STAMP 2.1.3 program to test for significant differences in pathway contributions [41]. A chi-square test corrected by Benjamini–Hochberg’s false discovery rate was applied. For graph building, data was used with a minimum relative abundance of 0.5 to determine proportion difference.

## 3. Results

### 3.1. Experimental Performance of Atlantic Salmon

Chloride cell movement was seen in the immune histochemistry analysis of Atlantic salmon subjected to the GSC, SS, and FD conditions, showing that the conditions were suitable for SW transfer (Appendix A). Additionally, RT-qPCR examination of ATPase-α and ATPase-β subunits validated this situation (Appendix A). In experimental groups, mortality was not observed.

### 3.2. Full-16S Sequencing Data Report

The Nanopore sequencing generated 5,867,910 reads filtered by Guppy and then processed for secondary metagenomic analysis. After demultiplexing, trimming, quality, and size-filtering, 46.18% of the reads were classified by NanoCLUST. A total of 96,160 reads were classified taxonomically, including the mock community, with a mean of 13,308 per condition. All non-classified bacteria were eliminated from the study; only classified taxa were included in the diversity and taxonomic studies (Appendix A).

### 3.3. Structure of The Intestinal Microbial Community

The rarefaction curves indicated that all samples reached the plateau, which means that the number of sequences was sufficient for taxonomic classification. However, only groups 10PSU-GSC and 20PSU-GSC had more than 10,000 reads (Appendix A). The Bray-Curtis dissimilarities for fish intestinal microbiota for all groups show values between 0.67–0.65. The transfer from FW to SW, including the gradient, shows a high discrepancy (0.68–0.91). The highest dissimilarity is observed at the first salinity change (10PSU-GSC: 0.91). Focusing only on the change from FW to 32PSU, we observe that 32PSU-GSC (0.67) presents the most minor dissimilarity, while 32PSU-SS and 32PSU-FD show a similar dissimilarity (0.88–0.85). Generally, the lowest dissimilarity was between the GSC groups (10PSU-GSC, 20PSU-GSC) and the groups 32PSU-SS and 32PSU-FD (Appendix A). The principal coordinate analysis (PCoA) using the Bray–Curtis dissimilarity showed four clusters. One grouped the 10PSU-GSC and 20PSU-GSC (Figure 1A). A second cluster grouped the 32PSU-SS and 32PSU-FD fish samples. The third cluster grouped the FW samples, and the fourth cluster consisted of the 32PSU-GSC independently distributed. The highest bacteria diversity was observed in individuals from group 10PSU-GSC, according to the Simpson (1-D) and Shannon indexes (Figure 1B,C), while the group with the least distribution was the FW. According to Pielou’s index, the group with the least uniformity was the intestinal fish at 20PSU-GSC (Figure 1D). Related to the samples obtained after one week in SW, the Simpson’s and Shannon’s indexes showed high bacterial diversity in fish from 32PSU-FD (Figure 1B,C). On the other hand, Pielou’s index showed more uniformity in group 32PSU-SS (Figure 1D).

### 3.4. Atlantic Salmon Intestine Microbiota Composition during The SW Transfer

All experimental groups identified 10 phyla in the intestinal microbiota (Figure 2A), and 89% of the microbiota were identified in *Proteobacteria phylum*. Additionally, the *Bacteroidetes*, *Firmicutes*, and *Actinobacteria* phyla were highly abundant in the microbiota sample. Intestine samples from FW were the only ones with the presence of *Armatimonadetes*. In contrast, the intestine microbiota from the 10PSU-GSC group had the presence of *Acidobacteria*, *Fusobacteria*, and *Verrucomicobia* (Figure 2A), whereas the phylum *Cyanobacteria* was distinguished in the intestinal microbiota of the 20PSU-GSC group. Moreover, the intestine microbiota of the 32PSU-SS group only exhibited bacteria of the phyla *Proteobacteria* and *Bacterioidetes*. Finally, the presence of *Thermotogae* phylum in the microbiota of the 32PSU-FD group was notable (Figure 2A).

The microbiota relative abundance of the top 25 genera showed a dominance of the genus *Escherichia-Shigella (Enterobacterales)* in all analyzed samples, except for the 10 PSU-GSC group. For this group, the specie distribution was more homogenous (Figure 2B). Notably, the intestine microbiota in the FW, 10 PSU, and 20 PSU samples exhibited more bacterial genera diversity than intestine microbiota at 32 PSU. Interestingly, the microbiota of samples of all groups at 32 PSU exhibited the presence of the genus *Vibrio*, with the highest abundance in the 32 PSU-SS group (Figure 2B).

Finally, a core microbiota was observed among all experimental groups. This core was composed of the orders *Flavobacteriales*, *Enterobacterales*, *Moraxellales*, and *Pseudomonadales* (Figure 3A). The analysis by genus reveals that the genus *Escherichia-Shigella* was the most abundant in the intestine core microbiota, followed by the genera *Acinetobacter* and *Pseudomonas* (Figure 3B).

### 3.5. Bacteria Species Identified in the Intestinal Atlantic Salmon Microbiota

Seventy clusters of species generated by NanoCLUST were above a threshold of the 98.7% of identity defined by Yarza et al. [14], equivalent to only 17.7% of the total clusters. The *Acinetobacter johnsonii* is the only specie that was identified in the intestinal microbiota of all experimental groups (Appendix A). In addition, species were observed that were presented in FW and maintained in the intestine microbiota of fish exposed to the gradual salinity change, such as *Pseudomonas migulae* and *Shigella flexneri*. Furthermore, the intestine microbiota of 32PSU-GSC contained species such as *Aliivibrio wodanis*, *Methylobacterium brachiatum*, *Microbacterium mangrove*, *Micrococcus luteus*, and *Sphingobium yanoikuyae*.

Through the NanoCLUST analysis, 46 clusters of potentially pathogenic bacteria were classified. However, only 5 represent an average identity and relative abundance above 98.7% and 0.2, respectively (Appendix A). These clusters of pathogenic groups identified in all data sets were the species *Acinetobacter johnsonii*, *Aliivibrio wodanis*, *Flavobacterium succinicans*, and *Providencia rettgeri*. Interestingly, the *Acinetobacter johnsonii* specie was present in the intestinal microbiota of all the experimental fish groups (Appendix A).

### 3.6. Intestine Microbiota after Seawater Transfer

In addition, the microbiota abundance and richness among fish groups after a week in seawater were evaluated. The phylum *Proteobacteria* was the most abundant in the intestine microbiota for each experimental fish group. Furthermore, this phylum exhibited the highest richness at the genus level. In particular, the genera *Vibrio*, *Pseudomonas*, *Escherichia-Shigella*, and *Acinetobacter* were present in the microbiota at 32PSU in all groups (Figure 4). The intestinal microbiota of the 32PSU-GSC group showed more richness of the phylum *Firmicutes*, especially the *Bacillales* and *Lactobacillales* genera. In addition, the 32PSU-SS group exhibited the highest abundance of the *Vibrio* genus, reaching 45%. The 32PSU-FD group showed more abundance and richness of the genera of the phylum *Bacteroidetes*. In addition, this group exhibited high richness of the phylum *Proteobacteria*, mainly the genus *Ralstonia* and *Pelomonas*, compared with the other groups at 32PSU (Figure 4). Finally, the 32PSU-FD group is the only one with the presence of the genus *Thermotogae* in its microbiota.

### 3.7. Functional Analysis in the Intestinal Atlantic Salmon Microbiota

We identified 338 functional pathways in the Atlantic salmon intestine microbiota with PICRUSt2 software [39]. Forty-eight of them were of level 2, and the majority are related to biosynthesis (Figure 5A, Appendix A). Pathways with an effect size > 0.5 and p < 0.05 among the salinity conditions were obtained by STAMP to compare the different SW treatments (Figure 5B–D). The most abundant pathways were associated with Biosynthesis of Vitamins, Amino Acids, Nucleoside and Nucleotide, Fatty acid, and Lipid (Figure 5A). The 5 pathways more abundant in the FW group were annotated at the level 2 descriptions Secondary Metabolite Degradation, Citrate cycle (TCA cycle), C1 Compound Utilization and Assimilation, and Nucleoside and Nucleotide Biosynthesis. Under the top 5 pathways more abundant in the 32 PSU-GSC group, TCA cycle was identified as a level 2 description. For the group 32 PSU-SS, Secondary Metabolic Degradation and TCA cycle were more abundant at the level 2 description, while the level 2 descriptions found with more abundance in the 32 PSU-FD group were Secondary Metabolite Degradation, Nucleoside and Nucleotide Biosynthesis, and TCA cycle (Appendix A).

In addition, the comparison among the FW and SW samples showed similar pathways. Notably, compared with SW, the FW metagenome showed a higher abundance of TCA cycle, Fatty Acid and Lipid Biosynthesis, Nucleoside and Nucleotide Biosynthesis, and Amino Acid Biosynthesis. The functional analysis revealed a significant difference in the metagenomics potential among the samples at 32 PSU. For instance, between 32 PSU-GSC and 32 PSU-SS, Aromatic Compound Degradation and TCA cycle were significantly higher in the 32 PSU-GSC group (Figure 5B). From the comparative analysis of 32 PSU-GSC and 32 PSU-FD, 12 pathways showed a difference in abundance, highlighting pathways such as Amino Acid Biosynthesis, Fatty Acid and Lipid Biosynthesis, and Amino Acid degradation, which exhibited high abundance in the 32 PSU-FD group (Figure 5C).

## 4. Discussion

Fish microbiota play a relevant role in the salmon’s performance and can be influenced by the diet, fish physiology, and environment, among other factors. During smoltification, Atlantic salmon are exposed to environmental change from freshwater (FW) to seawater [42]. Thus, their microbiota diversity can be modified and affect the seawater fish performance. Here, we used the full 16S rRNA sequencing to evaluate the Atlantic salmon intestinal microbiota variation during the SW transfer under different strategies. It was explored if a fish transfer to SW by gradual salinity change (GSC) exhibited differences in microbiota abundance and diversity compared with fish exposed to salinity shock (SS), as it is performed in the industry.

A total of 19 OTUs have been reported in the core intestine microbiota between FW and SW Atlantic salmon [20]. Among these OTUs, the authors reported the presence of *Lactobacillus*, *Lactococus*, *Streptococcus*, *Escherichia/Shigella*, *Pseudomonas*, and *Mycoplasma*. Moreover, Dehler et al. [26] reported *Escherichia/Shigella* in the core microbiota of Atlantic salmon pre-smolts in FW in aquarium and loch environment conditions. In addition, Lorgen-Ritchie et al. [9] describe the presence of *Pseudomonas* sp. in the core microbiota of FW/SW Atlantic salmon gut. Interestingly, this study identified a microbial core composed of *Pseudomonas*, *Escherichia/Shigella*, and *Acinetobacter*. The presence of *Acinetobacter* bacteria in Atlantic salmon gut microbiota has not been described previously. However, in FW Atlantic salmon, skin microbiota has been reported to have a high abundance of *Acinetobacter* [8]. It is suggested that the low diversity found in the intestine core microbiota is due to the study being carried out under laboratory conditions that can reduce the environmental microbiota.

Previously, the high species richness of Atlantic salmon gut microbiota in sea cages compared with FW gut microbiota has been reported [10]. Furthermore, increased bacteria diversity has been reported in the FW gut microbiota of Atlantic salmon in a RAS system compared with salmon gut microbiota transfer to SW [9]. In this study, the bacteria diversity analysis showed that the salmon intestinal microbiota of the 10PSU-GSC group exhibited the highest bacteria diversity among the intestine samples evaluated. Thus, this suggests that the gradual adaptation of salmon from FW to 10PSU-GSC favors the intestinal microbiota diversity in pre-smolts. Interestingly, among the intestine samples evaluated after a week in seawater, the Atlantic salmon microbiota exposed to GCS and the group feeding with the FD exhibited greater bacteria diversity than the group transferred to SW by salinity shock. Notably, the transcriptome analysis of the Atlantic salmon transferred to SW by gradual salinity change reported an expression increase of immune-related genes compared with salmon exposed to salinity shock [33]. Thus, it is possible to suggest that the increase of microbiota diversity in fish exposed to GSC can be associated with healthy fish. In general, low diversity in the intestinal microbiota is a marker of dysbiosis and has been associated with a wide range of diseases in humans [43]. Now, in fish, diseased-looking tilapia (ocular exophthalmos, skin hemorrhages, skin lesions, and necrosis) have been found to have a lower diversity index than healthy fish [44].

Among the phyla presented in the Atlantic salmon intestine microbiota, previous studies have reported a freshwater predominance of the phyla *Proteobacteria* and *Firmicutes* [9,10,21]. Additionally, high dominance of the *Proteobacteria* phylum has been reported in seawater [9]. Moreover, an increased presence of *Firmicutes* has been observed in the Atlantic salmon intestine after 20 weeks in SW [10]. Moreover, the phylum *Proteobacteria* has demonstrated an abundance decrease after eight months in seawater [10]. In this study, the phylum *Proteobacteria* was the most abundant in all the evaluated intestine samples. Among the treatments, *Firmicutes* and *Bacteroidetes* phyla were more abundant in the intestines of fish exposed to 10 and 20 PSU. Moreover, the *Firmicutes* phylum has been described in the gut core microbiota of Atlantic salmon identified between FW and SW [26]. Here, the GSC group’s intestine microbiota exhibited *Firmicutes* from FW to all salinity points. In addition, the intestine microbiota of the FD group presented the *Firmicutes* phylum. Thus, this suggests that the FD and the GSC stimulate the establishment of *Firmicutes*. Furthermore, the use of functional ingredients in salmon farming, such as insect meal, galactomannan oligosaccharides, and prebiotics, increases the Firmicutes abundance [3,10]. Notably, the Firmicutes phylum is not presented in the intestine microbiota of fish exposed to saline shock. Similarly, in wild salmon, the Firmicutes phylum is observed in stages associated with the FW, such as parr, smolts, and returning adults [45]. The gradual salinity change and the functional diet allow the presence of the *Firmicutes* phylum.

*Vibrio* genus was identified in the Atlantic salmon intestine microbiota after a week at 32 PSU, mainly in fish exposed to the salinity shock. Species of this genus are abundant in marine ecosystems [46,47]. For instance, *Vibrio* species have been reported in the intestines of shrimp, mollusks, abalones, and others [47]. Furthermore, the genus *Vibrio* is associated with pathogenicity in fish [48]. Additionally, after four weeks of SW transfer, the presence of the *Vibrionaceae* family in the gut microbiota of Atlantic salmon smolts has been reported [9]. Moreover, comparative study of the gut microbiota of Atlantic salmon in a control aquarium vs. a loch environment has indicated the presence of *Aliivibrio* in the core microbiota [20].

Notably, the genus *Lactobacillus* has been reported in greater abundance in Atlantic salmon intestine microbiota [9,10,26]. Moreover, the presence of *Lactobacillus* has been reported in the intestinal core microbiota of Atlantic salmon in FW and SW [26]. In addition, in a RAS system, the *Lactobacillus* species was identified among Atlantic salmon’s FW gut core microbiota [9]. In this study, the intestinal microbiota analyzed exhibited the absence of the genus *Lactobacillus*. However, *Streptococcus thermophilus*, from the *Lactobacillales* order, was identified in the intestine microbiota of 32 PSU-GSC. We hypothesize that the absence of the *Lactobacillus* genus in our study can be a consequence of the DNA extraction method used [49]. Another explanation can be associated with the Taq Polymerase used for the 16S library preparation, which may not favor the detection of this genus [50].

An advantage of using the complete 16S rRNA sequencing by Nanopore sequencing is the possibility to infer the microbiota metagenome. In this study, we used the V3–V4 region from 16S to perform functional analysis. However, the intestinal Atlantic salmon microbiota in this study showed some differences compared with other studies. For instance, in the FW microbiota of salmon, a high proportion of function associated with Carbohydrate Metabolisms, TCA cycle, Lipid Biosynthesis, Fatty Acid Biosynthesis, and oxidative phosphorylation has been reported [26]. However, in SW, fish microbiota exhibited a high enrichment of pathways such as Carbohydrate Metabolism, Amino Sugar and Nucleotide Sugar Metabolism, and Glycolysis/Gluconeogenesis. Additionally, differences in the metabolic processes of Atlantic salmon gut microbiota in FW and SW were reported by Lorgen-Ritchie et al. [9]. In this study, the intestine microbiota from SW groups exhibited a high relative abundance of pathways associated with Carbohydrate Degradation, Cell Structure, and Secondary Metabolite Degradation. Notably, in our study, the metagenomic analysis of FW and SW microbiota exhibited a high abundance of TCA cycle, Fatty Acid and Lipid Biosynthesis, Nucleoside and Nucleotide Biosynthesis, and Amino Acid Biosynthesis. In general, Amino Acid Biosynthesis stands out in metagenomic functional studies on intestinal fish [51]. Furthermore, valine, leucine, and isoleucine are essential metabolites for fish [52]. The 32 PSU-FD fish intestinal microbiota exhibited the highest enrichment of Amino Acid Biosynthesis. Contrary to our supposition, 32PSU-SS fish microbiota ranks second. However, focusing on the essential amino acid degradation pathway is more favored in the FD and SS treatments over GSC. Aromatic Compound Degradation is the pathway with the most significant differences between treatments. Aromatic amino acids, plant constituents, drugs, additives, colorants, and contaminants are some of the sources of aromatic compounds in the intestine [53]. Therefore, FD and GSC may have an advantage over SS in using aromatic compounds as a carbon source and eliminating toxins from the bacteria found in the fish intestine.

Concerning the findings archived during the study for the in-feed functional ingredients, there is strong evidence of the association of functional ingredients, such as essential fatty acids, nucleotides, yeast cell walls, and prebiotics, in the intestine fish microbiota modulation [10,54]. For instance, a high abundance of *Proteobacteria* and *Mycoplasmas* has been observed in Atlantic salmon feeding with functional ingredients after 44 weeks in seawater, compared with salmon feeding with a regular diet [10]. Moreover, a high difference in the microbiota diversity of sea bass (*Dicentrarchus Labrax*) feeding with functional ingredients (FI) has been reported [54]. Furthermore, sea bass feeding with FI reduces potentially pathogenic bacteria, such as *Vibrionales* [54]. Notably, the results obtained in the current study revealed that fish feeding with FD exhibited a higher bacteria genus richness than the other experimental groups. Furthermore, Atlantic salmon feeding with FD at 32 PSU showed a low abundance of *Vibrio* genus compared with the other experimental groups. Thus, besides functional ingredients improving the fish immune system, it also impacts the fish microbiota abundance and richness.

Our results show the potentiality of salmon microbiota studies to determine the smolt condition. It was demonstrated that the gradual salinity change increases the salmon microbiota abundance, thus improving their health. The implementation of an Atlantic salmon transfer by gradual exposure to seawater could increase the transfer’s success. Nowadays, the salmon industry has been increasing the time of fish in the RAS conditions before transferring to seawater as a strategy to reduce the exposure to sea pathogens [55]. Thus, the industry could implement the gradual salinity change in RAS systems, improving salmon performance in seawater.

## 5. Conclusions

The differences in the diversity indices demonstrate variation in the intestinal microbiota of Atlantic salmon during the transfer from FW to SW. The richness and diversity indices were more significant in the intestinal microbiota of Atlantic salmon exposed to a gradual salinity change. *Proteobacteria* were dominant in all groups, but there were variations in lower taxonomic levels in all conditions. *Escherichia/Shigella* were found in high abundance as part of SW microbiota. Despite the taxonomic differences found among treatments, a low percentage of the metabolic pathways analyzed showed significant differences; most of them are related to biosynthesis. Finally, this is the first study that evaluates the intestinal microbiota of Atlantic salmon exposed to gradual salinity change during smoltification, and it shows that this way of transferring to SW increases the diversity of the microbiota of the fish and, thus, improves their health condition.

## Figures and Tables

**Figure 1 microorganisms-11-00076-f001:**
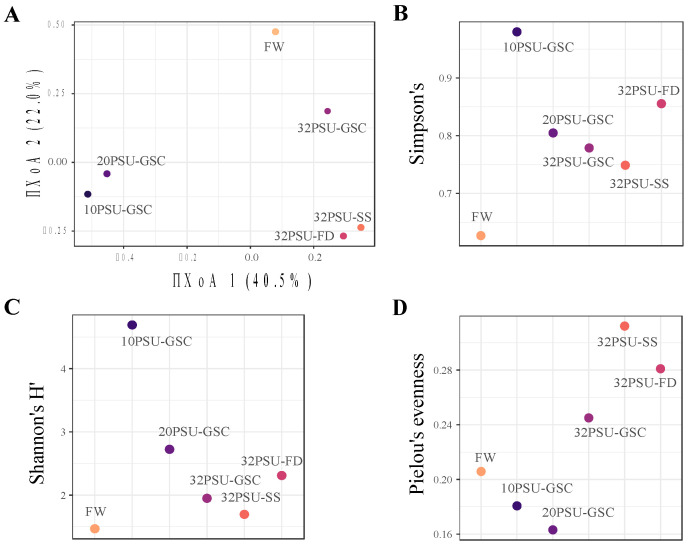
Diversity index of Atlantic Salmon intestinal microbiota between salinity treatments. FW: Fresh water previous treatment group, GSC: gradual salinity change at 10 PSU, 20 PSU, and 32 PSU groups; 32PSU-SS Salinity shock at 32-PSU group; 32PSU-FD: Salinity shock at 32 PSU group previously feeding with a functional diet. (**A**) PCoA plot of the beta diversity of microbiomes calculated by Bray-Curtis dissimilarity; (**B**) The Simpson’s 1-D; (**C**) Shannon diversity index; and (**D**) Pielou’s evenness.

**Figure 2 microorganisms-11-00076-f002:**
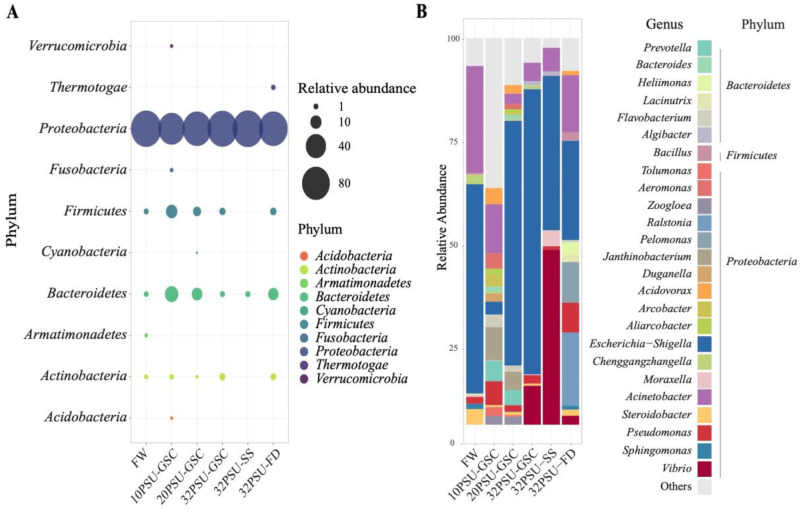
Relative abundance of intestinal Atlantic Salmon microbiota composition. FW: Fresh water previous treatment group; GSC: gradual salinity change at 10 PSU, 20 PSU, and 32 PSU groups; 32PSU-SS Salinity shock at 32-PSU group; 32PSU-FD: Salinity shock at 32 PSU group previously feeding with a functional diet. (**A**) microbiota abundance at the phylum level; (**B**) Top 25 most abundant taxa at genus level between treatments.

**Figure 3 microorganisms-11-00076-f003:**
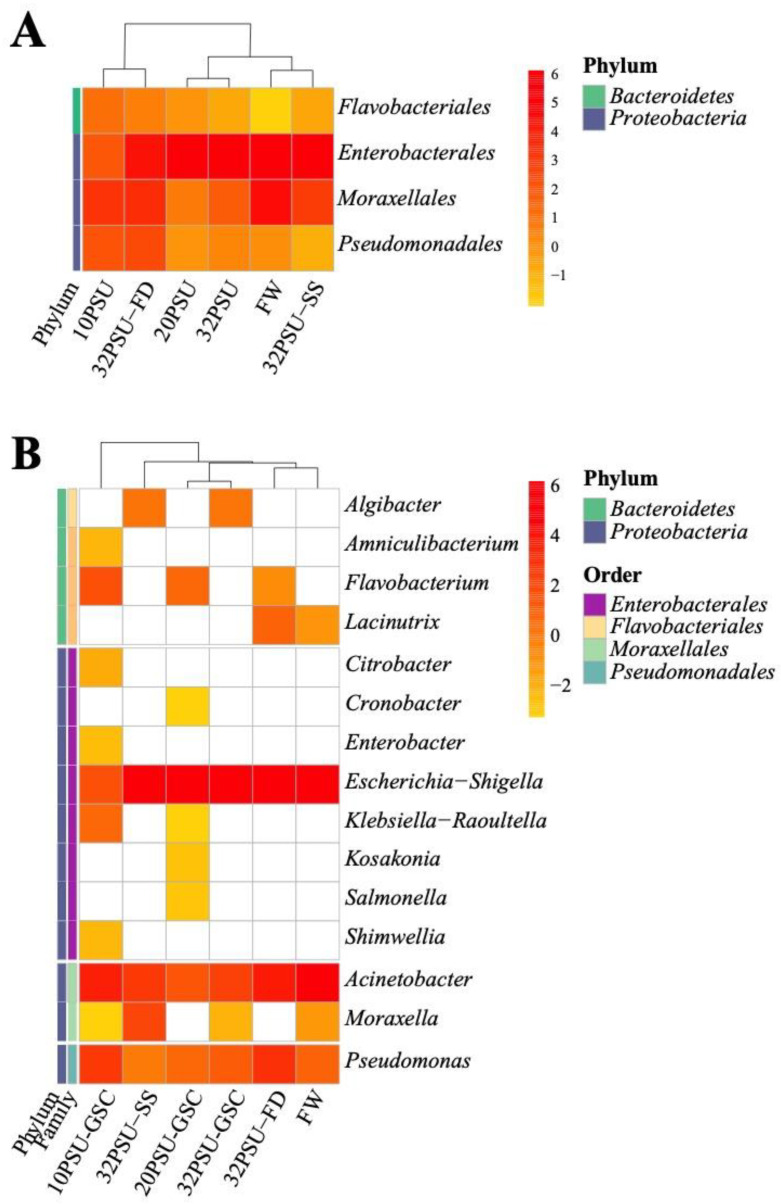
Heat map of normalized relative abundances of the Atlantic salmon intestine core microbiota among all experimental groups. FW: Fresh water previous treatment group; GSC: gradual salinity change at 10 PSU, 20 PSU, and 32 PSU groups; 32 PSU-SS Salinity shock at 32-PSU group; 32 PSU-FD: Salinity shock at 32 PSU group previously feeding with a functional diet. (**A**) Core microbiota at the order level; (**B**) Core microbiota composition at the genus level.

**Figure 4 microorganisms-11-00076-f004:**
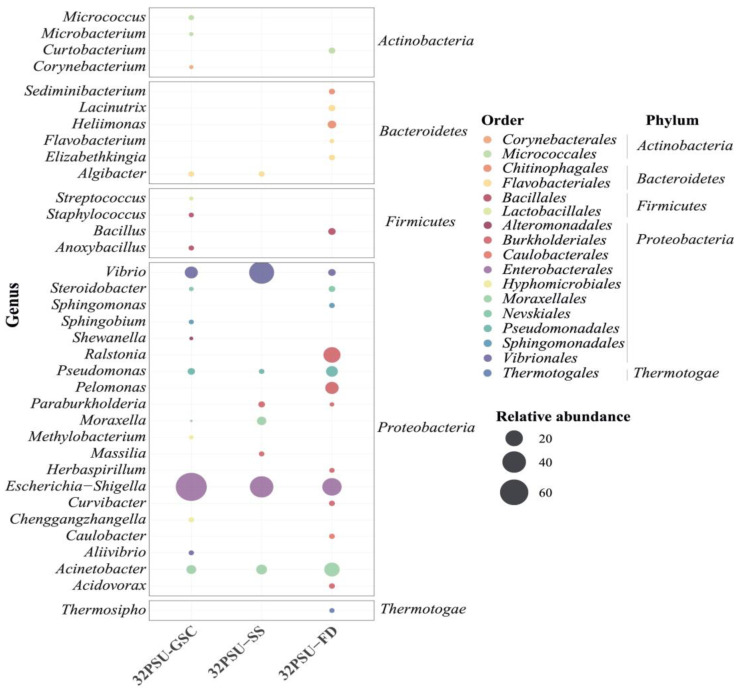
Relative abundance of intestinal Atlantic Salmon microbiota composition at the genus level in seawater groups. 32PSU-GSC: gradual salinity change at 32 PSU group; 32 PSU-SS Salinity shock at 32-PSU group; 32 PSU-FD: Salinity shock at 32 PSU group previously feeding with a functional diet.

**Figure 5 microorganisms-11-00076-f005:**
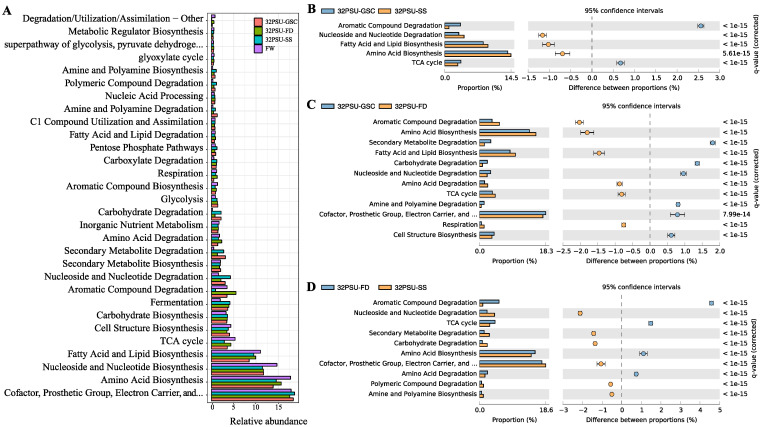
Relative abundance proportion of metabolic pathway (level 2). FW: Fresh water previous treatment group, GSC: gradual salinity change at 10 PSU, 20 PSU, and 32 PSU groups; 32 PSU-SS Salinity shock at 32-PSU group; 32 PSU-FD: Salinity shock at 32 PSU group previously feeding with a functional diet. (**A**) Metabolic pathways abundance in freshwater (FW) and different seawater treatment; (**B**–**D**) STAMP analysis of level 2 metabolic pathways with different abundance among samples at 32 PSU. PICRUSt2 inferred metabolic pathways with a filter size of 0.5.

## Data Availability

Not applicable.

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
