# Peer review of "Atlantic Salmon (Salmo salar) Transfer to Seawater by Gradual Salinity Changes Exhibited an Increase in The Intestinal Microbial Abundance and Richness"

_microorganisms, 2022, doi:10.3390/microorganisms11010076_

Round 1

Reviewer 1 Report

General comments:

I would like to thank the authors for their work and willingness to contribute to the research field. My comments bellow were made with the sole intention of helping to improve the final article.

After reading your work, I have some general and specific questions, suggestions or comments that I outline bellow.

The major issue I have with this work is the experimental design. For the GSC treatment we have fish that were manipulated at least 3 times for intestine samples and fish that were kept in Treatment conditions for a month. For SS we have fish that were only manipulated once and were only under treatment conditions for one week. And for FD we have fish that were in a pre-treatment conditions for a month followed by a week under treatment with different manipulations again. This is not a similar design, treatments vary in more that what they are supposed to be testing, meaning that the control is not a complete true control. How did the authors manage this? This is not described in the methods. The impacts of this, on the results and on the results interpretation, is not acknowledged nor discussed.

From Fig 1, it seems that there is also and FW control. Is this correct? It is not mentioned on the methods. How were manipulations controlled for in FW? How long were fish kept? When were samples taken?

What happen to fish after the experiments? This was not described in the methods, and it should?

Specific comments:

Line 15 – I suggest to replace the word “can” with the word “may”.

Line 26 – “adaptation of Atlantic salmon in seawater” – Adaptation in sea water to what, are fish adapting to what? Furthermore, the term adaptation should be used with extreme care. Is the adaptation from the fish? Is it from the microbial community? And, in this case isn’t it community adaptation trough species and individuals’ selection?

Line 57 – The species scientific name should be italicized. Furthermore, to all scientific species names in the text should be done the same.

Line 78 – Please write “GO” in full.

Figure 1 caption it is not self-explanatory. Please write FW in full In the caption and all the other acronyms and abbreviations as well.

Figure 2 – All acronyms for the treatments must be written in full in the caption.

Figure 3 - All acronyms for the treatments must be written in full in the caption.

Table 1 - All acronyms for the treatments must be written in full in the caption.

Figure 4 - All acronyms for the treatments must be written in full in the caption.

Figure 5 - All acronyms for the treatments must be written in full in the caption.

Lines 339-341 – “nterestingly, among the intestine samples evaluated after a week in seawater, the Atlantic salmon microbiota exposed to GCS and feeding with a functional diet” – From the methods I do not find this double treatment.

Lines 359 – “adaptation enhancer” – please elaborate and/or describe what is meant by this expression. Additionally, support with valid literature.

Line 364 – For the diadromous portion of the population? Please clarify in the text.

In Supllentary figures and tables - All acronyms and abbreviations must be written in full in the caption.

Reviewer 2 Report

Comments:

Obviously, the authors did a complete and excellent work. Three treatments were used to investigate the response of intestinal microbiome to different salinity scenarios. The results found that gradual salinity change or functional diet pre-feeding may be beneficial for the activity of intestinal microbiota, thereby improving the adaption of salmon during seawater transfer. Overall, the writing is good and the points are clear. I have some tiny suggestions or queries that may help to improve the quality of this manuscript.

1.       Lines 100-101, what’s the composition of the functional diet? Is that a business secret? If so, please clarify that why you choose this diet for seawater transfer. Is that widely used by salmon farmers in Chile?

2.       Table 1 can be listed in the Supplementary Materials.

3.       In the Discussion, could you please provide more evidence about the improvement on the metabolism/activity of intestinal microbiota by gradual salinity change? In Lines 390-398, you only mentioned that there are great variations in energy metabolism or synthesis of biological macromolecules between the freshwater and seawater fish microbiota. Please give more clarifications.

4.       Similar to the above comments, could you give more discussions on how the functional diet worked and benefitted for the metabolism of microbiota (Lines 410-415)? Any evidence from the metabolic pathways?

5.       I strongly suggest the authors to add some perspectives on the applications of gradual salinity change or functional diet in salmon farming. You need to highlight the importance of the work, e.g. the possibility to improve fish health condition, reduce the cost of farming…
